# Tuning magnetic spirals beyond room temperature with chemical disorder

Mickaël Morin[1], Emmanuel Canévet[2], Adrien Raynaud[1], Marek Bartkowiak[1], Denis Sheptyakov[2], Voraksmy Ban[3], Michel Kenzelmann[1], Ekaterina Pomjakushina[1], Kazimierz Conder[1] & Marisa Medarde[1]

In the past years, magnetism-driven ferroelectricity and gigantic magnetoelectric effects have been reported for a number of frustrated magnets featuring ordered spiral magnetic phases. Such materials are of high-current interest due to their potential for spintronics and low-power magnetoelectric devices. However, their low-magnetic ordering temperatures (typically $<100\,K$) greatly restrict their fields of application. Here we demonstrate that the onset temperature of the spiral phase in the perovskite $YBaCuFeO_5$ can be increased by more than $150\,K$ through a controlled manipulation of the Fe/Cu chemical disorder. Moreover, we show that this novel mechanism can stabilize the magnetic spiral state of $YBaCuFeO_5$ above the symbolic value of $25\,°C$ at zero magnetic field. Our findings demonstrate that the properties of magnetic spirals, including its wavelength and stability range, can be engineered through the control of chemical disorder, offering a great potential for the design of materials with magnetoelectric properties beyond room temperature.

[1] Laboratory for Scientific Developments and Novel Materials, Paul Scherrer Institut (PSI), CH-5232 Villigen, Switzerland. [2] Laboratory for Neutron Scattering and Imaging, Paul Scherrer Institut (PSI), CH-5232 Villigen, Switzerland. [3] Swiss Light Source, Paul Scherrer Institut (PSI), CH-5232 Villigen, Switzerland. Correspondence and requests for materials should be addressed to Ma.M. (email: marisa.medarde@psi.ch).

Spiral magnetic order in non-geometrically frustrated transition metal oxides often arises from the competition between nearest-neighbour $J$ and next-nearest-neighbour (NNN) $J'$ superexchange interactions[1]. As a true competition may only happen if the two contenders are of comparable strengths, $J$ and $J'$ should not be too different in magnitude. Yet in insulating transition-metal oxides NNN interactions are usually very weak[2]. A direct consequence is that magnetic spirals usually appear in the presence of small nearest-neighbour couplings, leading naturally to low-order temperatures. There are very few examples of materials where both $J$ and $J'$ reach values large enough to stabilize spiral phases close to room temperature. One of them is CuO, which combines a large $J$ (ref. 3) with a $J'/J$ ratio $\sim -1/5$ (ref. 4) close to the spiral stability criterion for a frustrated one-dimensional (1D) spin chain (Supplementary Fig. 1)[1]. A spiral phase is observed for temperatures above 200 K, but only in a narrow temperature window of 17 K (ref. 5). Different strategies aimed to engineer the spiral stability range towards higher temperatures were proposed, but they turned out to be either unsuccessful[6] or unpractical for applications[7]. In view of the high current interest on spiral magnets as potential for low-power magnetoelectric devices[8–20], it is highly desirable to identify new mechanisms that yield high-temperature spiral phases in oxides.

In the following, we address this question by investigating the layered perovskite YBaCuFeO$_5$, one of the few oxides with a spin-spiral phase above 200 K (refs 5,20,21). As for CuO[22], spontaneous electric polarization was reported to exist at zero magnetic field in the spiral phase[21,23–24], which is stable over a temperature range more than 10 times larger than in cupric oxide. We show that the Cu/Fe chemical disorder in YBaCuFeO$_5$ has a tremendous impact on the degree of magnetic frustration, and that it can be used to tune the stability range of the spiral phase and to extend it beyond room temperature.

## Results

**Magnetic transitions.** The crystal structure of YBaCuFeO$_5$ (space group $P4\,mm$), is shown in Fig. 1a[21]. It consists of two perovskite units where the A-site cations $Ba^{2+}$ and $Y^{3+}$ are ordered in layers. The B sites host $Cu^{2+}$ and $Fe^{3+}$, but their tendency to order is less pronounced because of their similar ionic size. 1/6 of ordered O vacancies reduce the coordination of the B-site from octahedral to square-pyramidal, and as a result the BO$_5$ units, which are connected by the apexes, form layers of 'bow ties' stacked along the **c** axis.

YBaCuFeO$_5$ undergoes two magnetic transitions at temperatures $T_{N1}$ and $T_{N2}$ (refs 21,23–28). The high temperature collinear antiferromagnetic (AFM) phase $(T_{N1} > T > T_{N2})$ is characterized by the propagation vector $\mathbf{k_c} = (\frac{1}{2}\ \frac{1}{2}\ \frac{1}{2})$ (Fig. 1b). In the **ab** plane all possible nearest-neighbour $J_{ab}$ interactions (Cu–Cu, Cu–Fe and Fe–Fe) are AFM and very strong (up to 130 meV), whereas weaker, up to two orders of magnitude smaller AFM ($J_{c1}$) and ferromagnetic (FM) $J_{c2}$ couplings alternate along the **c** axis[21]. The only FM coupling in the structure corresponds to ions occupying the bow-ties ($J_{c2}$, Fig. 1a), and, according to the Goodenough-Kanamori-Anderson rules[29–31], this is only possible if they are occupied by Cu/Fe pairs. Since Cu and Fe are usually disordered in this material, this implies that the bow-ties are preferentially occupied by FM Cu–Fe 'dimers' randomly distributed in the structure. The lower ground state energy of Fe–Cu distributions containing disordered FM Cu–Fe dimers has been confirmed by density functional theory (DFT) calculations[21].

Below $T_{N2}$ the magnetic order becomes incommensurate along the **c** axis and the neutron powder diffraction (NPD) data are consistent with the appearance of an inclined circular helix with propagation vector $\mathbf{k_i} = (\frac{1}{2}\ \frac{1}{2}\ \frac{1}{2}\ \pm\ q)$ (ref. 21). Interestingly, the deviation from collinearity in the spiral state occurs exclusively within the bow-ties, which lose their FM alignment below $T_{N2}$ (Fig. 1c). This suggests that $J_{c2}$ is the magnetic coupling most sensitive to the thermal variation of the crystal structure. Other external perturbations such as high pressure, isovalent A-site substitutions or variations in the Fe/Cu distribution can thus be expected to affect it also significantly. To check this hypothesis, we prepared five YBaCuFeO$_5$ ceramic samples using identical conditions apart from the cooling speed after the last annealing , which was systematically varied in order to obtain distinct degrees of Cu/Fe disorder (see Methods).

Figure 1d shows the magnetic susceptibility $M/H$ of the five samples, where the two magnetic anomalies $T_{N1}$ and $T_{N2}$ are clearly visible. While $T_{N1}$ decreases slightly with increasing cooling rates, $T_{N2}$ displays the opposite behaviour and undergoes a huge increase of more than 150 K, reaching $T_{N2} = 310$ K for the sample with the fastest cooling rate. The appearance of Bragg reflections corresponding to $\mathbf{k_c} = (\frac{1}{2}\ \frac{1}{2}\ \frac{1}{2})$ below $T_{N1}$ in the neutron diffraction patterns (Fig. 1e) confirms that this transition corresponds to the onset of the collinear AFM order. Below $T_{N2}$ the magnetic reflections are progressively replaced by new incommensurate satellites corresponding to $\mathbf{k_i} = (\frac{1}{2}\ \frac{1}{2}\ \frac{1}{2}\ \pm\ q)$, characteristic of the magnetic spiral state (Supplementary Fig. 2). The temperature dependence of the magnetic modulation vector $q$ is shown in Fig. 2a for the five samples and the ground state value ($q_G$) is shown in Fig. 2b as a function of cooling rate. The angle $\varphi_G$ between the **ab** plane and the rotation plane of the spiral at 10 K is also displayed in Fig. 2b. Interestingly, there is a clear correlation between the cooling rate, the collinear-to-spiral transition temperature $T_{N2}$ and the ground state values of $q$ ($q_G$) and $\varphi$ ($\varphi_G$). This means that the three main properties of the spiral, namely, the stability range, the periodicity and the inclination of the rotation plane can be tuned in a controlled way by the choice of the synthesis protocol.

**Crystal structure.** To get insight about the mechanism behind these large magnetic differences, we employed high-resolution neutron and synchrotron X-ray powder diffraction (see Methods and Supplementary Fig. 3) to investigate the structural modifications induced by the different cooling rates. The possible role of oxygen non-stoichiometry was ruled out after determining the total O-content by thermogravimetric H$_2$-reduction, which was found to be in excellent agreement with the nominal chemical formula and with the results from the Rietveld analysis (see Methods). Figure 3a shows the evolution of the room temperature lattice parameters relative to the values obtained for the sample with the slowest cooling rate. A small increase of $a$ ($+0.02\%$) and a decrease of $c$ about three times larger ($-0.06\%$) are observed for faster cooling speeds. This means that a tensile ($c/2a < 1$) tetragonal distortion results in a more robust spiral state, which grows at the expense of the collinear phase (Fig. 3b). Figure 3c,d show the thickness of the bipyramid block ($d_2$) and that of the layers separating the bipyramids along $c$ ($d_1$). While $d_1$ undergoes only a modest increase ($+0.005$ Å), the decrease in $d_2$ is twice larger ($-0.01$ Å). The large shrinkage of $d_2$ is thus at the origin of the contraction of $c$ that is observed for faster cooling rates.

Figure 4a,b show the evolution of the average apical and basal distances for the FeO$_5$ and CuO$_5$ pyramids. As expected from its Jahn-Teller electronic configuration ($t_{2g}^6\ e_g^3$), the coordination polyhedron of $Cu^{2+}$ is strongly distorted, with shorter in-plane distances and a much longer apical distance than $Fe^{3+}$ ($t_{2g}^3\ e_g^2$). Interestingly, none of the Fe–O/Cu–O distances change significantly with the cooling rate within the experimental error.

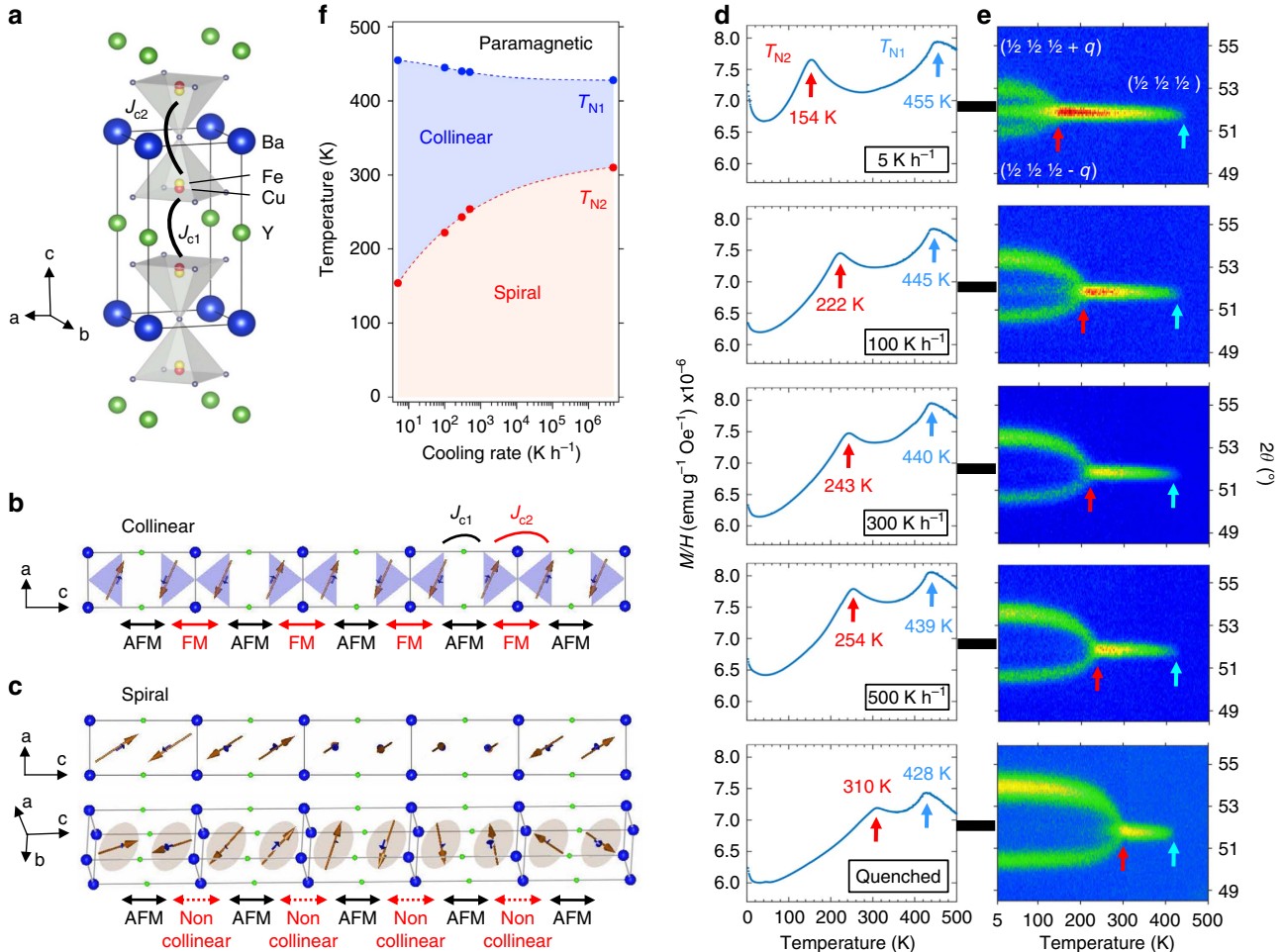

**Figure 1 | Dependence of the magnetic transition temperatures $T_{N1}$ and $T_{N2}$ with the cooling rate.** (**a**) Crystal structure of YBaCuFeO$_5$ showing the Cu/Fe disorder in the bipyramidal sites. (**b,c**) Magnetic structures in the commensurate collinear (**b**) and incommensurate spiral phases (**c**). In both cases, only $\frac{1}{4}$ of the magnetic unit cell is shown. The rotation plane of the spiral is indicated for clarity. The nearest-neighbour magnetic couplings along the **c** crystal axis are also shown. (**d**) Mass magnetic susceptibility $M/H$ as a function of temperature measured under the application of an external magnetic field of 5,000 Oe for the five YBaCuFeO$_5$ powder samples prepared using different cooling rates in the last annealing. (**e**) Contour maps showing the temperature dependence of the position and the intensities of the magnetic Bragg reflection (½ ½ ½) associated to the high-temperature collinear antiferromagnetic phase, and the (½ ½ ½ $\pm$ $q$) satellites of the low-temperature magnetic spiral phase for the five samples. The measurements were performed by heating at the neutron powder diffractometer DMC (SINQ, Switzerland) using a wavelength $\lambda = 4.5\,\text{Å}$. The blue and red arrows indicate, respectively, the magnetic transition temperatures $T_{N1}$ and $T_{N2}$. (**f**) Stability range of the collinear and spiral phases with the cooling rate. Dashed lines are guides for the eye.

This suggests that the contraction of the bow-ties may arise from a modification of the Cu/Fe distribution induced by different cooling speeds.

To check this scenario we examined the Cu/Fe occupation of the split B-sites. Their values, averaged over the whole sample volume, are shown in Fig. 4c,d and Supplementary Table 1 as function of the cooling rate. We note that the smallest difference between the Cu and Fe occupations (10%) is obtained for the quenched sample, where the Cu/Fe disorder promoted by the enhanced ionic diffusion at high temperatures is 'frozen' by the fast cooling rate. Slower cooling results in a clear tendency towards a more ordered state reflected by a larger difference between the Cu/Fe occupations, which reaches 16% for the sample cooled at 5 Kh$^{-1}$. The large degree of Cu/Fe disorder in all samples is supported by the values of the mean-square displacements (MSD) at 10 K. As shown in Fig. 4e,f and Supplementary Table 1, they are very small for Cu, Fe, Y and the basal oxygens, both in the **ab** plane and along the **c** axis, consistent with the reduced thermal motion expected at this temperature. For Ba and the apical oxygen, the MSD are also

small in the **ab** plane, but very large in the **c** direction, signaling a static contribution $<u_S^2> \sim 0.016$–$0.022\,\text{Å}^2$ superimposed to the very weak thermal motion $<u_T^2> \sim 0.002$–$0.005\,\text{Å}^2$. This reflects the length fluctuations of the apical distances in the bipyramids[32,33], confirming the large degree of Cu/Fe disorder in all samples.

## Discussion

A remarkable observation is that very small differences in the average Cu/Fe disorder have a dramatic impact in the degree of magnetic frustration, resulting into a huge increase of the spiral stability range. A full understanding of this behaviour needs additional information on the details of the Cu/Fe distribution that can provided by techniques such as scanning microscopies, extended X-ray absorption fine structure (EXAFS) or diffuse scattering. However, the observed evolution of $d_1$ and $d_2$ can already provide important hints about the expected impact of disorder on the couplings $J_{c1}$ and $J_{c2}$. Looking to Figs 1f and 3c,d and Supplementary Fig. 4, we note that $T_{N1}$ and $1/d_1$ decrease for

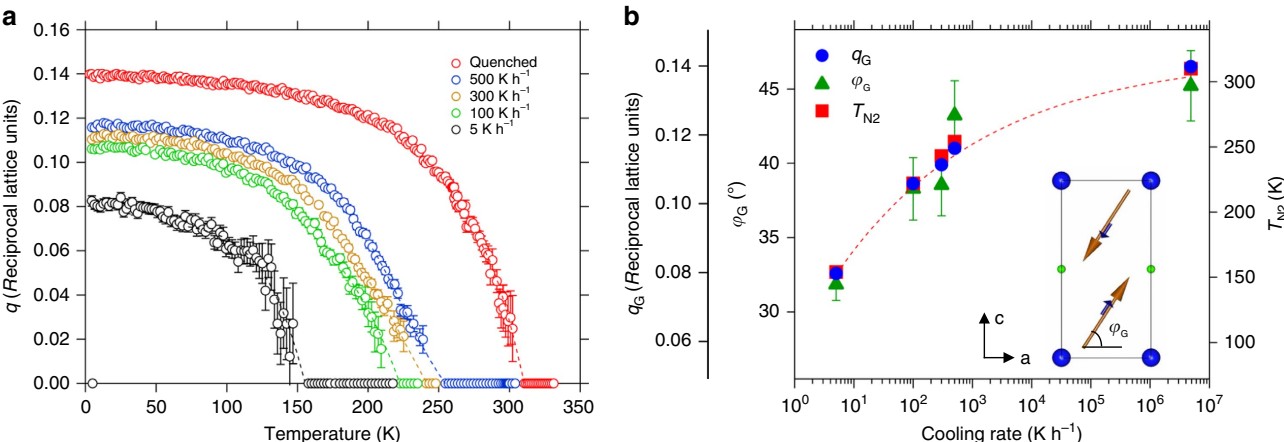

**Figure 2 | Changes in the spiral magnetic order with the cooling rate.** (**a**) Temperature dependence of the modulus of the magnetic modulation vector $q$ for the five $YBaCuFeO_5$ samples. For each sample, $q$ has been set to zero for $T > T_{N2}$. The dashed lines are guides for the eye. (**b**) Evolution of the collinear-to-spiral transition temperature ($T_{N2}$) and the ground state values ($T = 10\,K$) of the modulus of the magnetic modulation vector ($q_G$) and the inclination of the spiral rotation plane ($\varphi_G$). The figure shows the positive correlation between the three quantities and the cooling rate. The error bars of $q_G$ and $T_{N2}$ are smaller than the size of the symbols and the dotted line is a guide for the eye. The inclination angle $\varphi$ is the complementary of the angle $\theta = 90 - \varphi$ used in ref. 21. The error bars of $q$(**a**) and $q_G$ and $\varphi_G$ (**b**) are the standard deviations obtained from the fits of the magnetic structure using the FullProf Suite Rietveld package[41,42].

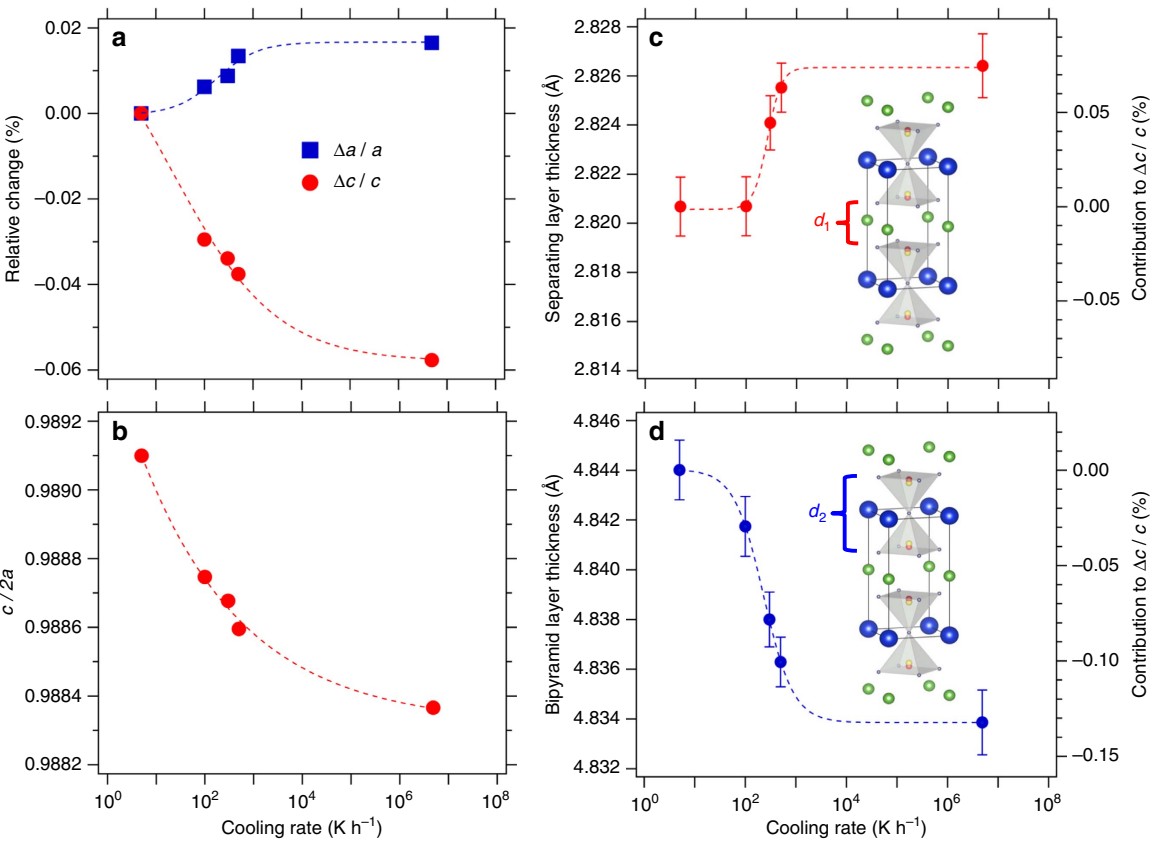

**Figure 3 | Changes in the room temperature crystal structure with cooling rate.** (**a**) Relative change (%) of the lattice parameters $a$ and $c$ with respect to those of the sample with the lowest $T_{N2}$. (**b**) Tensile ($c/2a < 1$) tetragonal distortion. (**c**) Thickness of the Y-containing separating layers ($d_1$). (**d**) Thickness of the bipyramid layers ($d_2$). The right vertical axis in **c** and **d** indicates the percent contribution of $d_1$ and $d_2$ to the variation of the **c** axis with the cooling rate. Dashed lines are guides for the eye. All values were extracted from the combined (neutron and X-ray synchrotron) Rietveld fits of the powder diffraction data at room temperature. The error bars of the lattice parameters and the interatomic distances are the standard deviations obtained from the fits of the crystal structure using the FullProf Suite Rietveld package[41,42]. In the case of **a** and **b**, they are smaller than the marker size.

increasing cooling rates whereas $T_{N2}$ and $1/d_2$ display the opposite behaviour. It is thus tempting to associate the control of $T_{N1}$ (and hence the appearance of 3D order) to inter-bipyramid coupling $J_{c1}$, and the set-up of the spiral state at $T_{N2}$ to the intra-bipyramid coupling $J_{c2}$. Also, given that the increase of $T_{N2}$ with the cooling rate is much faster than the decrease of $T_{N1}$, the $J_{c2}/J_{c1}$ ratio is probably a more appropriate control parameter.

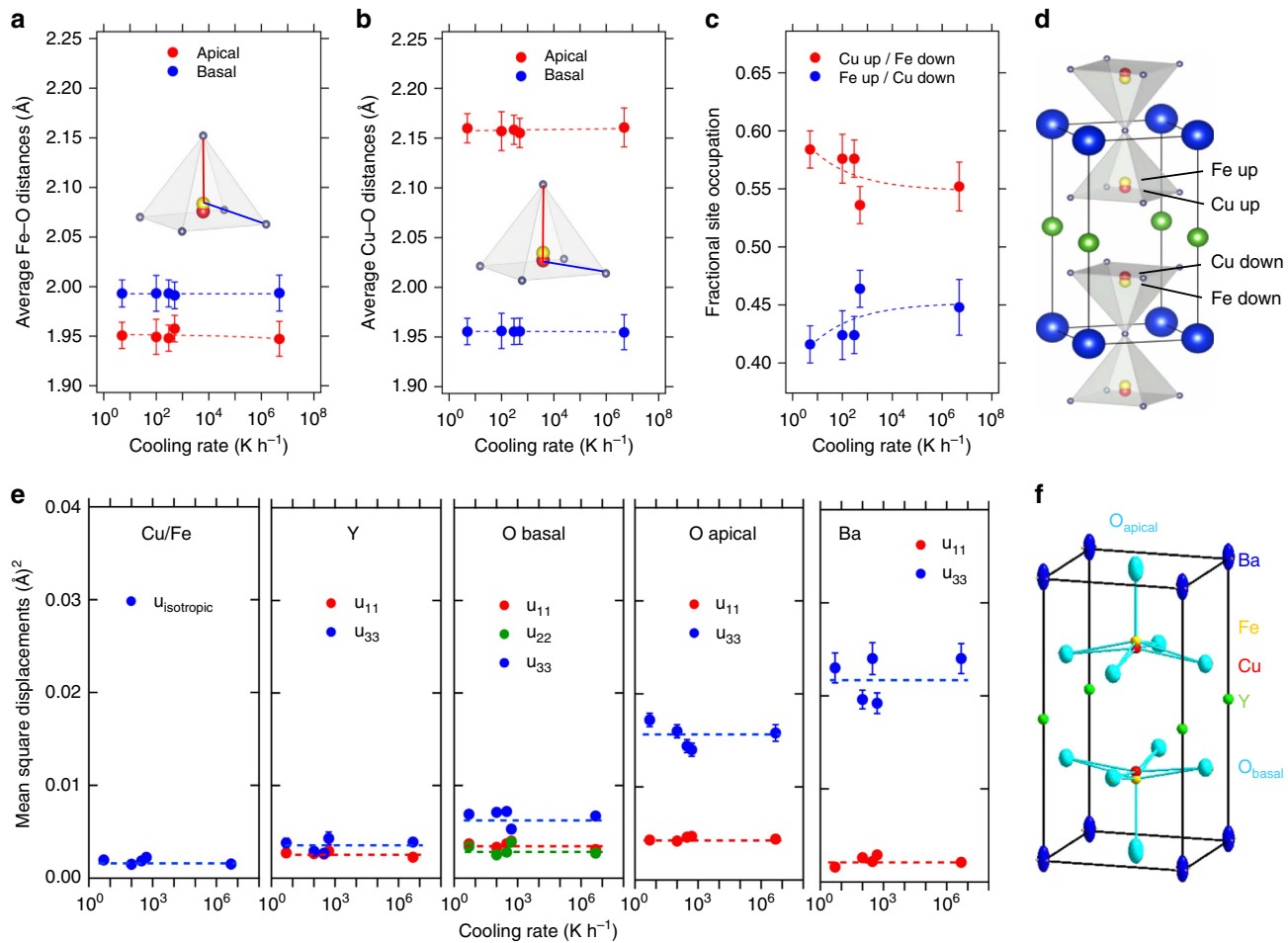

**Figure 4 | Evolution of the average Cu/Fe chemical disorder with the cooling rate.** (**a,b**) Fe–O and Cu–O interatomic distances at room temperature (RT). (**c–d**) Cu/Fe occupation of the split B-sites in the pyramids, as obtained from the combined neutron and X-ray synchrotron Rietveld fits at RT. (**e**) Anisotropic mean-square displacements (MSD's) of all atomic sites at 10 K. (**f**) Thermal ellipsoids (90% probability) at RT. Dashed lines are guides for the eye. The error bars of the interatomic distances, the Cu/Fe occupation and the MSD's are the standard deviations obtained from the fits of the crystal structure using the FullProf Suite Rietveld package[42,43].

The empirical relationship $T_{N2} \propto J_{c2}/J_{c1}$ suggests that the spiral magnetic order in YBaCuFeO$_5$ could display some of the characteristics of a quasi-1D chain. However, the alternating AFM-FM-AFM-FM… chain along the **c** axis suggested by the collinear magnetic order (Fig. 1b) is not frustrated[34]. As shown in ref. 21, competition between nearest-neighbour ($J_{c1}$ and $J_{c2}$) and NNN couplings along **c** could give rise to a spiral. However, the NNN couplings obtained by *ab initio* DFT calculations either are too small, or had the wrong sign[21]. The origin of the magnetic frustration in YBaCuFeO$_5$ remains thus mysterious, indicating that some important ingredient is still missing.

Since the degree of average Fe/Cu disorder has a prominent role in the control of $T_{N2}$ and $q_G$, disorder-based frustration models could provide additional insight. Although the idea that disorder could promote order is counter-intuitive, it has been shown in the past that site and/or bond defects in a magnetically ordered lattice may, in some particular conditions, give rise to a spiral[35,36]. If a disorder-based mechanism is at the origin of the spiral in YBaCuFeO$_5$, the most affected coupling will probably be the intra-bowtie coupling $J_{c2}$ because both, its sign and its magnitude are strongly dependent of which cations occupy the bowties (Cu–Fe: FM, Fe–Fe: AFM, Cu–Cu: negligible small). Interestingly, it is found experimentally that the deviation from collinearity by entering the spiral phase occurs exclusively within the bipyramidal units (Fig. 1b,c).

As mentioned in introduction, the bowties are expected to be occupied by FM Cu–Fe pairs[21]. On the other hand, we show in Fig. 3c,d that the size of the pyramids ($d_2$) changes with the degree of average disorder, suggesting that the 'Cu–Fe occupation' rule may not be always fulfilled. Based on the results of DFT calculations we do not expect a lot of Fe–Fe/Cu–Cu defects (as shown in ref. 21, Cu/Fe distributions with Fe–Fe and/or Cu–Cu pairs in the bipyramids are more expensive). However, we cannot exclude having small, cooling-rate dependent amounts. Interestingly, Fe–Fe defects are strongly AFM (about 100 times larger in absolute value than the FM Cu–Fe coupling). Hence, a few of them could produce important perturbations in the underlying collinear magnetic order. Establishing whether they may turn the collinear order into a spiral is out of the scope of this work and will need further investigation.

To summarize, we have shown that Cu–Fe chemical disorder has a prominent role in the control of the temperature stability range of the two-ordered magnetic phases present in YBaCuFeO$_5$. Although disorder changes very little the energy scale responsible for the paramagnetic-to-collinear AFM transition at $T_{N1}$, it dramatically increases the degree of frustration and the related collinear-to-spiral transition at $T_{N2}$. As a result, the stability range of the spiral phase is extended by $>150$ K and its upper limit pushed beyond room temperature at zero magnetic field. Our findings show that this novel mechanism can be effectively used

to engineer the properties of a magnetic spiral, including its wavelength, orientation and stability range. Although our samples are too leaky to sustain electric polarization at room temperature, these results may be relevant for the design of other spiral magnets with magnetoelectric properties beyond room temperature.

## Methods

**Synthesis.** The $YBaCuFeO_5$ ceramic samples were prepared by solid state synthesis. High purity (Aldrich, 99.999% trace metals basis) stoichiometric amounts of $BaCO_3$, $Y_2O_3$, $CuO$ and $Fe_2O_3$ were used to prepare 40 g of starting material. After a pre-annealing of $Y_2O_3$ oxide at 900 °C for 10 h the starting oxides were thoroughly mixed and fired at 1,150 °C for 50 h under oxygen gas flow. The obtained black powder was grounded again and divided into five identical portions that were pelletized, separately sintered at 1,150 °C for 50 h in air and cooled to room temperature in different conditions. We used 5, 100, 300 and 500 K h$^{-1}$ for four of the samples, whereas the fifth one was quenched into liquid nitrogen. The phase purity was checked by laboratory X-ray powder diffraction (Brucker D8 Advance, Cu $K\alpha$), which indicated the absence of impurity phases and an excellent crystallinity. The oxygen content, as determined from thermogravimetric $H_2$-reduction, was very close to the sample formula. All samples showed deviations from the nominal stoichiometry smaller than 1%, in agreement with the results of the Rietveld analysis.

**Magnetic susceptibility.** DC magnetization measurements were carried out on a superconducting quantum interference device magnetometer (MPMS XL, Quantum Design) equipped with oven. $YBaCuFeO_5$ pellets ($m \sim 20$ mg, $D \sim 3$ mm, $H \sim 1$ mm) from the same batches as the samples used for the neutron and X-ray diffraction measurements were mounted in transparent drinking straws and cooled in zero field down to 1.8 K. The magnetization M of the sample was then measured in a magnetic field $B = \mu_0 H = 0.5$ T up to 400 K by heating. For the high-temperature measurements (300–500 K) the samples were wrapped in Al foil as described in ref. 37. After application of a magnetic field of 0.5 T the magnetization was measured by heating. The signal from the empty sample holders was separately measured in the same conditions and subtracted from the data. The magnetic susceptibility $\chi^{DC} = M/B$ was then calculated for all samples. The values of $T_{N1}$ and $T_{N2}$ mentioned in the text correspond to the maxima of the $\chi^{DC}$ vs temperature curves.

**Neutron and synchrotron X-ray diffraction.** NPD measurements were carried out at the Swiss Neutron Source SINQ of the Paul Scherrer Institute in Villigen, Switzerland. The samples were introduced in cylindrical vanadium sample cans ($D = 0.6$ cm, $H = 5$ cm) and mounted on the stick of a cryofurnace. Neutron diffraction patterns were continuously recorded at the powder diffractometer DMC[38,39] (Pyrolitic Graphite (002), $2\theta_{max} = 104°$, $2\theta_{step} = 0.1°$, $\lambda = 4.5$ Å), while ramping the temperature from 1.5 to 500 K. Longer acquisitions for magnetic structure refinements were made at 10 K and room temperature with $2\theta_{max} = 130°$. High-resolution patterns at these two temperatures were also recorded at the powder diffractometer HRPT[40] (Ge (822), $2\theta_{max} = 160°$, $2\theta_{step} = 0.05°$, $\lambda = 1.1546$ Å). In both instruments the background from the sample environment was minimized using oscillating radial collimators. The wavelengths and zero offsets were determined using a NAC reference powder sample. The values of $T_{N1}$ and $T_{N2}$, defined, respectively, as the set-up and the maximum of the (½ ½ ½) magnetic Bragg reflection, were found to coincide with maxima of the $\chi^{DC}$ vs temperature curves (Supplementary Fig. 2).

Synchrotron X-ray powder diffraction measurements were performed at the Swiss Light Source (SLS) of the Paul Scherrer Institute in Villigen, Switzerland. All samples were loaded in borosilicate glass capillaries ($D = 0.1$ mm, $\mu R = 0.53$) and measured at room temperature in transmission mode with a rotational speed of $\sim 2$ Hz at the Materials Science Beamline[41] (Si (111), $\lambda = 0.77627$ Å). The primary beam was vertically focused and slitted to about $300 \times 4,000 \, \mu m^2$. Powder diffraction patterns were recorded at eight different detector positions for 10 s at room temperature using a Mythen II 1D multistrip detector (Dectris) with energy discrimination ($2\theta_{max} = 120°$, $2\theta_{step} = 0.0036°$, threshold at 12,000 eV) and then binned into one pattern. The wavelength and zero offset were determined using a Si reference powder sample (NIST SRM 640d).

**Data analysis.** All diffraction data were analysed using the Rietveld package FullProf Suite[42,43]. The structural and magnetic refinements were carried out by combining the data sets recorded at the same temperature: room temperature (HRPT + MSBL, Supplementary Fig. 3); 10 K (HRPT + DMC, Supplementary Fig. 2). We used the non-centrosymmetric space group $P4\,mm$ for the description of the crystal structure, which enables to refine separately the z coordinates and the occupation of the split Cu and Fe sites. Anisotropic Debye-Waller factors were used for all atoms with exception of Cu and Fe. The z coordinates of the two basal oxygen sites O2 and O2′ were refined separately but their MSD were restricted to have the same value (Supplementary Table 1). The possible existence of extra oxygen in the Y layers was checked by introducing a new O site at the position (½, ½, z) with $z \sim 0.5$. Attempts to refine the z coordinate and the site occupancy leaded to unstable fits, indicating that extra oxygen, if any, is below the detection limit of NPD.

The collinear and spiral magnetic structures were described according to the models reported in ref. 44. The ratio between the Fe and Cu magnetic moments was restricted to have the same ratio than their free ion, spin-only values (5:1). Below $T_{N2}$, were the collinear and spiral phases coexist (Supplementary Fig. 2 and Supplementary Table 1), the Fe and Cu magnetic moments were restricted to have the same value and the same inclination with respect to the **ab** plane in the two magnetic phases.

**Data availability.** Raw powder diffraction data were generated at the SINQ and SLS facilities (Paul Scherrer Institut, Switzerland). Derived data supporting the findings of this study are available from the corresponding author.

**Code availability.** FullProf Suite is available free of charge at https://www.ill.eu/sites/fullprof/

**Note added after acceptance of the manuscript.** Shortly before acceptance of the manuscript for publication in Nature Communications, two theoretical papers by A. Scaramucci and co-workers became available in ArXiv (ArXiv 1610.00783 and 1610.00784). These companion works present a random magnetic exchange-based model which, for certain kinds of chemical disorder, can give rise to magnetic spirals stable at high temperatures. Such model predicts a constant, disorder-independent $T_{spiral}/k_G$ ratio, which is nicely fulfilled by the five samples presented in our manuscript (using the $T_{N2}$ and $k_G = 0.5$—$q_G$ values of Fig. 2b, it is straightforward to see that there is a linear relationship between these two quantities with a slope $T_{N2}/k_G = -2,616$ (92)). The spiral state in $YBaCuFeO_5$ could thus be an experimental realization of such model. The two ArXiv papers have been added to the reference list[44,45].

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

## Acknowledgements

We thank A. Scaramucci, N. Spaldin, M. Müller, C. Mudry, T. Shang, R. Frison and H-B. Bürgi for fruitful discussions. This work was supported by the Swiss National Science Foundation (Grant No. 200021-141334) and the European Community's 7th Framework Program (Grant No. 290605; COFUND: PSI-FELLOW). We acknowledge the allocation of beam time at the Swiss Neutron Source SINQ (HRPT and DMC diffractometers) and the Swiss Light Source SLS (Materials Science Beamline).

## Author contributions

Mi.M. and Ma.M. conceived and led the project, Mi.M., A.R., K.C. and E.P. synthesized the samples, Mi.M., Ma.M., A.R., E.C. and D.S. performed the neutron diffraction measurements, Mi.M. and V.B. carried out the synchrotron X-ray diffraction experiments, Mi.M., A.R. and M.B. measured the magnetic susceptibility, Mi.M., Ma.M., A.R. and E.C. analysed all the experimental data. Ma.M., Mi.K. and Mi.M. wrote the paper with the input from all authors.

## Additional information

**Competing financial interests:** The authors declare no competing financial interests.

