## [Peer Review File · Nature Communications]

Reviewers' Comments:

Reviewer #1 (Remarks to the Author):

Report on the manuscript NCOMMS-16-17266-T "Tuning magnetic spirals beyond room temperature with chemical disorder" by M. Morin et al.

This manuscript reports a remarkable enhancement of the temperature for the appearance of a spiral spin structure from 150 K for a slowly cooled sample to above 300 K for a quenched sample of YBaCuFeO₅. The temperature for the onset of a collinear antiferromagnetic structure is weakly suppressed from 455 K to 428 K for the same samples.

The change of the temperature for the formation of a spiral spin structure is accompanied by changes of the lattice parameters and certain layer distances (measured at room temperature) and Cu/Fe occupation of the split B-sites.

The reported results are of special interest since the incommensurate spin order is accompanied by electric polarization and magneto-electric coupling, with an onset temperature that is tunable and can reach above room temperature.

The manuscript is worthy of publication in Nature Communication.

A couple of unclear points that should be amended:

The last sentence of the upper paragraph of page 6 "We note also that an extremely slow cooling rate would fully suppress the spiral state ($T_{N2}=0$)." This statement is not supported by the experiments and should be removed.

Mid paragraph page 7: "Interestingly, any of the Fe-O/Cu-O distances change significantly with the cooling rate within the experimental error." Should read: "Interestingly, not any of the Fe-O/Cu-O distances....."

Bottom page 8: "It is thus tempting to associate the control of T_{N2} (and hence the appearance of 3D order) to inter-bipyramid coupling J_{c2} and the setup of the spiral state at T_{N1} to the intra-bipyramid coupling J_{c1} ." The labelling in this sentence is mixed-up both as to the transition temperatures: T_{N1} should refer to the onset of long range collinear order and T_{N2} to the spiral ordering; as well as to J_{c1} (which refers to the inter bi-pyramid) and J_{c2} (which labels the intra bi-pyramid coupling).

Reviewer #2 (Remarks to the Author):

The authors present a study demonstrating a dramatic impact of Cu/Fe disorder on the stability of a spin-spiral magnetic phase in YBaCuFeO₅, by combining macroscopic data and structural analysis on samples cooled with different speeds. The data are of high quality and the above conclusion follows unambiguously. As the authors write, additional information on the details of the Cu/Fe disorder will be necessary to obtain a full understanding of how this proceeds, but in the discussion they outline a very plausible mechanism. I think their finding is novel and important, far beyond the specific material considered. It is presented clearly and the manuscript is very interesting to read. The paper deserves publication, but only after the following two concerns are addressed.

1. The link between the bow-tie contraction and i) Cu/Fe disorder in general as well as ii) the preservation of the preferential occupation of the bipyramidal units by FM Fe-Cu pairs is not very clear and should be explained better. It might be useful to compare with optimized distances in the various models calculated in earlier DFT work.
2. In the discussion, the authors note correlations between the increase or decrease of the two magnetic transition temperatures and distances. However, in logarithmic scales, the change in distances (Fig. 3c/d) appear rather abrupt, while the Neel temperatures seem to evolve more

smoothly (in particular largest change between the lowest and second-lowest cooling rate rather than at higher cooling rates). Should the evolution of the Neel temperatures not be roughly proportional to the evolution of distances?

In addition I have a few minor suggestions for the authors:

- In Figs. 3c/d it would be useful to have as a second vertical scale a normalized length so that the different percentage change is immediately visible.
- It might be better to exchange the labeling of Jc1 and Jc2 so that the numbers are consistent with the correlating Neel temperatures and distances.
- Information of how the question of the detailed distribution of Fe and Cu (when partially ordered) could be addressed in future work would be useful.

Reviewer #3 (Remarks to the Author):

In this experimental study, the authors demonstrated that stabilities, critical temperatures and wave-vectors of a spiral spin order in a perovskite-type magnet YBaCuFeO5 can be tuned by controlling the extent of chemical disorder through manipulating the cooling rate. They showed that the critical temperature of spiral spin order in this material can be enlarged by 150 K and reaches 310 K by tuning the chemical disorder.

It is known that spiral spin orders in magnetic insulators often give rise to magnetoelectric effects via inducing ferroelectric polarization, which is currently attracting intensive research interest from the viewpoints of both fundamental science and technical applications. However, spiral spin orders caused by magnetic frustration between ferromagnetic and antiferromagnetic interactions tend to have low critical temperatures or small stability domain because the frustration inevitably reduces a magnetic energy scale. The low critical temperatures of spiral-magnetism-based multiferroic materials have been a critical problem to be solved for their device applications.

In this sense, the finding that a room-temperature spin spiral order can be achieved by chemical disorder is important and interesting, and may provide a novel concept to this field. However, I found that there are a lot of critical points which should be addressed as follows.

(1) Argument in the introduction part is hard to understand. The authors claim that spiral magnetic order often arises from competition between next-neighbor J and other higher-order J'. However, to my knowledge, the competition which stabilizes the spiral order is usually that between nearest-neighbor exchange and next-nearest-neighbor exchange. The authors mentioned CuO as a material which shows a high-temperature magnetic spiral phase, and claim that its origin is large J and J' together with a J'/J ratio $\sim -1/5$. The authors should give more elaborate explanations with a schematic figure of spatial configuration of exchange interactions in CuO.

(2) The physical origin of the spiral spin order in YBaCuFeO5 is hard to understand. Why the spiral order is stabilized within the authors' model with Jc1 and Jc2? The configuration of Jc1 and Jc2 shown in Fig.1b has no frustration irrespective of the sign of Jc2, and hence cannot be expected to cause a spiral spin order.

(3) A physical mechanism for the disorder-induced enhancement of critical temperature is argued in the text. The authors claim that contraction of the bow-ties along the c-axis may arise from modification of the Cu/Fe distribution induced by different cooling speed. This fact seems to provide a key to understanding a physical mechanism for the observed phenomenon, but it is totally hard to understand.

First, why the modification of Cu/Fe distribution causes the contraction of the bow-ties ?, and how the strengthened Jc2 on the shortened bow-ties contributes to stability of the spiral spin order ?

When the extent of disorder in Cu and Fe distributions increases, randomness in the spatial configuration of ferromagnetic and antiferromagnetic Jc2 bonds (bow-ties bonds) should increase. We naturally expect that this randomness should cause instability of the coherent spiral spin order because ordered arrangement of frustrated J1 and J2 exchanges is advantageous for the coherent spiral spin order. But the authors' finding is that the randomness makes the spiral order more stable. Why such a counterintuitive thing happens?

(4) The experimental data for ferroelectric polarization is not shown. The authors demonstrated that the chemical disorder stabilizes the spiral spin order. I guess that the disorder, however, also makes the insulating property of the sample worse and renders the sample leaky. If it is true, the sample cannot remain ferroelectric at high temperatures even though the spin spiral survives. Because the authors argue a potential for design of room-temperature magnetoelectrics, they should argue the ferroelectric polarization at high temperatures or the upper limit of temperature that the ferroelectric polarization can be measured.

(5) If possible, the authors should try to argue effects of annealing on the insulating properties. The annealing can be expected to improve the insulating properties and hence the magnetoelectric properties.

NCOMMS-16-17266-T

Answers to Reviewer #1

We would like to thank the referee for her/his positive appraisal of our manuscript and for recommending its publication in Nature Communications. In the new, revised version, we have taken into account his/her comments and suggestions. The modifications can be summarized as follow:

The last sentence of the upper paragraph of page 6 “We note also that an extremely slow cooling rate would fully suppress the spiral state ($TN_2=0$).” This statement is not supported by the experiments and should be removed.

- The paragraph containing this statement in page 6 has been removed.

Mid paragraph page 7: “Interestingly, any of the Fe-O/Cu-O distances change significantly with the cooling rate within the experimental error.” Should read: “Interestingly, not any of the Fe-O/Cu-O distances.....”

- This sentence has been reformulated in the sense indicated by the referee.

Bottom page 8: “It is thus tempting to associate the control of TN_2 (and hence the appearance of 3D order) to inter-bipyramid coupling J_{c2} and the setup of the spiral state at TN_1 to the intra-bipyramid coupling J_{c1} .” The labelling in this sentence is mixed-up both as to the transition temperatures: TN_1 should refer to the onset of long range collinear order and TN_2 to the spiral ordering; as well as to J_{c1} (which refers to the inter bi-pyramid) and J_{c2} (which labels the intra bi-pyramid coupling).

- The typos in the labeling of the NN exchange constants along the c axis have been corrected. As the referee points out, the subindex 1 should refer to the parameters related to the setup of the collinear magnetic order and the subindex 2 to those related to the spiral magnetic order.

Answers to Reviewer #2

We thank the referee for the positive assessment of our work, as well as for recommending publication in Nature Communications. Her/his comments and suggestions have been incorporated to the new version of the manuscript. We summarize the changes below together with the answer to the referee’s questions and comments:

1. The link between the bow-tie contraction and i) Cu/Fe disorder in general as well as ii) the preservation of the preferential occupation of the bipyramidal units by FM Fe-Cu pairs is not very clear and should be explained better. It might be useful to compare with optimized distances in the various models calculated in earlier DFT work.

The only information about the disorder that can be obtained from diffraction techniques is the Cu/Fe occupation of the pyramids averaged over the whole sample, and this point is mentioned in the text. However, we showed in ref. 21 (*Phys. Rev. B* **91**, 064408-064421 (2015)) that the sign of the coupling within the bipyramidal units is strongly dependent of the Cu/Fe occupation (Fe-Fe: AFM; Cu-Cu: negligibly small; Cu-Fe: FM). Since the coupling within such units in the collinear phase is FM (see Fig. 1b), we concluded that the bowties should be mainly occupied by Cu-Fe pairs. Such “dimers” have to be disordered in order to be consistent with the large average Cu-Fe disorder observed experimentally.

In Fig. 3d of the present manuscript we show that the size of the bipyramids (d_2) changes monotonically with the cooling rate (and hence with the degree of average disorder). This suggests that these units are not always occupied by Cu-Fe pairs. Moreover, it indicates that the number of Fe-Fe / Cu-Cu “defects” changes monotonically with the cooling rate. Because the number of Cu/Fe sites is constant, it is not easy to predict how this would affect d_1 and d_2 . The DFT calculations for the different occupation models of ref. 21 were unfortunately not conclusive.

The reasons supporting the Cu-Fe preferential occupation of the bowties are presented at the end of page 4. The link between the change in d_2 and the existence of Fe-Fe/Cu-Cu defects is discussed in the first paragraph of page 9, which has been modified in the revised version. We hope that the arguments are now presented in a clearer and more convincing way.

2. In the discussion, the authors note correlations between the increase or decrease of the two magnetic transition temperatures and distances. However, in logarithmic scales, the change in distances (Fig. 3c/d) appear rather abrupt, while the Neel temperatures seem to evolve more smoothly (in particular largest change between the lowest and second-lowest cooling rate rather than at higher cooling rates). Should the evolution of the Neel temperatures not be roughly proportional to the evolution of distances?

- In Figs. 3c/d it would be useful to have as a second vertical scale a normalized length so that the different percentage change is immediately visible.

The Neel temperatures are (in general) roughly proportional to the exchange interactions, The exchange interactions are, however, not directly proportional the interatomic distances. This is because in insulators, the J 's are directly related to the overlap integrals, which decrease by increasing distances in a complex way which depends on the lattice geometry. As shown in the new Supplementary Figure 4, we find that in YBaCuFeO_5 T_{N1} and T_{N2} are roughly proportional to $1/d_1$ and $1/d_2$, respectively.

A new vertical scale with the percentage of c axis-change contributed by d_1 and d_2 has been added to Figs. 3c/d as suggested by the referee.

- It might be better to exchange the labeling of J_{c1} and J_{c2} so that the numbers are consistent with the correlating Neel temperatures and distances.

Referee 1 noted also the typos with the 1-2 labeling. They have been now corrected all along the paper. “1” refers to the parameters related to the setup of the collinear magnetic order, and “2” to those related to the spiral magnetic order.

- Information of how the question of the detailed distribution of Fe and Cu (when partially ordered) could be addressed in future work would be useful.

To take into account this point we have added a sentence in the first paragraph of page 8 with possible techniques which may provide information about the details on the Cu-Fe distribution in the structure.

Answers to Reviewer #3:

We would like to thank the referee for the positive assessment concerning the interest and the importance of our work in the context of multiferroic-related research. We have revised our manuscript according to her/his comments and suggestions. We summarize the changes below together with our answers to the referee’s questions and criticisms:

(1) Argument in the introduction part is hard to understand. The authors claim that spiral magnetic order often arises from competition between next-neighbor J and other higher-order J' . However, to my knowledge, the competition which stabilizes the spiral order is usually that between nearest-neighbor exchange and next-nearest-neighbor exchange.

The expression “higher orders” was used in the introduction as a synonym of “next-nearest” (i.e., second, third or even fourth next-nearest neighbors). We agree with the referee in that this choice is inexact and may lead to confusion. The first paragraph of the introduction has been modified according to the referee’s remark. We hope that the reasons at the origin of the low order temperatures observed in spiral magnets are now conveyed in a clearer way.

The authors mentioned CuO as a material which shows a high-temperature magnetic spiral phase, and claim that its origin is large J and J' together with a J'/J ratio $\sim -1/5$. The authors should give more elaborate explanations with a schematic figure of spatial configuration of exchange interactions in CuO.

A new figure illustrating the geometry of the quasi-1D chains in CuO and showing the main exchange interaction has been included in the Supplementary Information. According to the calculations of refs. 4 and 7, one of the 2 superexchange couplings along the chain axis ($J_z \sim 80-108$ meV, AFM) is the largest one in the structure. The other coupling along the chain (J_x) is much smaller and FM. All the other NN / NNN couplings are at least 4-5 times smaller than J_z in absolute value.

(2) The physical origin of the spiral spin order in YBaCuFeO5 is hard to understand. Why the spiral order is stabilized within the authors' model with J_{c1} and J_{c2} ? The configuration of J_{c1} and J_{c2} shown in Fig.1b has no frustration irrespective of the sign of J_{c2} , and hence cannot be expected to cause a spiral spin order.

(3) A physical mechanism for the disorder-induced enhancement of critical temperature is argued in the text. The authors claim that contraction of the bow-ties along the c-axis may arise from modification of the Cu/Fe distribution induced by different cooling speed. This fact seems to provide a key to understanding a physical mechanism for the observed phenomenon, but it is totally hard to understand.

First, why the modification of Cu/Fe distribution causes the contraction of the bow-ties ?, and how the strengthened J_{c2} on the shortened bow-ties contributes to stability of the spiral spin order ?

When the extent of disorder in Cu and Fe distributions increases, randomness in the spatial configuration of ferromagnetic and antiferromagnetic J_{c2} bonds (bow-ties bonds) should increase. We naturally expect that this randomness should cause instability of the coherent spiral spin order because ordered arrangement of frustrated J_1 and J_2 exchanges is advantageous for the coherent spiral spin order. But the authors' finding is that the randomness makes the spiral order more stable. Why such a counterintuitive thing happens?

We thank the referee for this series of excellent questions. We will try to answer them as a block.

In our previous study (ref. 21), we showed that NNN couplings along c are most probably not at the origin of the spiral. As the referee points out, the collinear magnetic order (Fig. 2b) is not frustrated, so the origin of the spiral remained unexplained. Our new manuscript provides a new important experimental information, namely, that T_{N2} seems to be controlled by $J_2/J_1 \propto d_1/d_2$. This finding is **purely** empirical and the only thing that suggests is that the microscopic model at the origin of the spiral has some “quasi-1D” characteristics. The fact that the alternating AFM-FM-AFM-FM chain along the c axis suggested by the collinear magnetic order of Fig. 1b is not frustrated shows nevertheless that some important ingredient is still missing.

Since chemical disorder plays a clear role in the control of T_{N2} and q_G , models based in the existence of frustration due to disorder could provide some additional insight. As pointed out by the referee, the fact that disorder could promote order is counter-intuitive, but it has been shown in the past that site and/or bond defects in a magnetically ordered lattice may give rise to a different ordered state –in particular a spiral-. (Phys. Rev. Lett. **62**, 1564 (1989). Phys. Rev. B. **53**, 2633 (1996).

If a disorder-based mechanism is at the origin of the spiral in YBaCuFeO5, the most affected coupling will probably be the intra-bowtie coupling J_{c2} because both, its sign and its magnitude are strongly dependent of which cations occupy the bowties (Cu-Fe: AFM,

Fe-Fe: AFM, Cu-Cu: negligible small). Indeed, the deviation from collinearity by entering the spiral phase occurs *exclusively* within the bipyramidal units (Figs 1b and c). As mentioned in the answer to referee 2, the bowties are expected to be occupied by FM Cu-Fe pairs. However, in view of the monotonic changes in the bowtie size d_2 with increasing average Fe/Cu disorder we cannot exclude having small, cooling rate-dependent amounts of Fe-Fe/Cu-Cu “defects”. Interestingly, Fe-Fe defects are strongly AFM (about 100 times larger in absolute value than the FM Cu-Fe coupling). Hence, a few of them could produce important perturbations in the underlying collinear magnetic order. In order to test this hypothesis our theoretician colleagues at the ETH Zurich and the PSI are presently investigating whether the presence of such defects in a lattice with the YBaCuFeO₅ topology may give rise to a spin spiral.

The question about the link between average Fe/Cu disorder and the changes in the interatomic distances d_1 and d_2 was already addressed in our answer to Referee 2. In Fig. 3d of the present manuscript we show that the size of the bipyramids (d_2) changes monotonically with the cooling rate (and hence with the degree of average disorder). This suggests that these units are not always occupied by Cu-Fe pairs. Based in our previous DFT calculations we don't expect a lot of Fe-Fe/Cu-Cu defects (as shown in ref. 21, Cu/Fe distributions with double occupation in the bipyramids are more expensive). However, we cannot exclude having small, cooling-rate dependent amounts. If this is the case, small changes in the bowties size (d_2) can be expected, which, due to the relaxation of the structure, will result in modifications of other distances (for example d_1). Because the number of Cu/Fe sites is constant, it is not easy to predict how this would affect d_1 and d_2 . The DFT calculations for the different occupation models of ref. 21 were unfortunately not conclusive. The question remains thus open and will have to be addressed in future studies.

In order to make all these points clear the “DISCUSSION” section has been completely rewritten and two additional references added (35 and 36).

(4) The experimental data for ferroelectric polarization is not shown. The authors demonstrated that the chemical disorder stabilizes the spiral spin order. I guess that the disorder, however, also makes the insulating property of the sample worse and renders the sample leaky. If it is true, the sample cannot remain ferroelectric at high temperatures even though the spin spiral survives. Because the authors argue a potential for design of room-temperature magnetoelectrics, they should argue the ferroelectric polarization at high temperatures or the upper limit of temperature that the ferroelectric polarization can be measured.

(5) If possible, the authors should try to argue effects of annealing on the insulating properties. The annealing can be expected to improve the insulating properties and hence the magnetoelectric properties.

Our samples, not compact enough for transport measurements (70-80%), are indeed too leaky to sustain polarization at RT. Low pellet densities favor the creation of trapped charges which result in artifacts during polarization measurements. This makes difficult

to provide temperature below of which polarization can be measured. As the referee suggests, longer annealing may improve the situation. We are presently exploring this possibility, as well as other routes aimed to increase the pellet density.

Following the Referee's recommendation this point is now briefly mentioned at the end of the manuscript (page 9). Also, the final statement about the possibility of using these results for the design of high temperature magnetoelectrics has been reformulated in a milder way.

Reviewers' Comments:

Reviewer #2 (Remarks to the Author):

In the resubmitted manuscript, the authors have made extensive changes in response to the points raised by myself and the other two referees. In particular, they have rewritten much of the discussion, mainly in response to criticism of the third referee. The new discussion section is much clearer, particularly in regard of what is directly shown, what is reasonably surmised, and what yet remains to be done to further clarify the origin of their experimental discovery.

With their point-by-point answer to the referee reports, the changes in the manuscript, and additional supplementary information, the authors in my opinion adequately address the concerns raised in my previous report, as well as those raised by the other referees.

Hence, I recommend publication of the paper.

I have only minor suggestion to check start of second paragraph of the discussion (what is supposed to be proportional appears no consistent with first paragraph).

Reviewer #3 (Remarks to the Author):

I found that all the criticism and comments I raised in the previous report have been satisfactorily addressed the revised manuscript is now suitable for publication in Nature Communications.

NCOMMS-16-17266B

Answers to Reviewer #2

We would like to thank the referee for recommending publication of the revised manuscript in Nature Communications, as well as by noting the inconsistency in the paragraph 2 of the discussion.

I have only minor suggestion to check start of second paragraph of the discussion (what is supposed to be proportional appears no consistent with first paragraph).

- This has been corrected in the final version. Following the discussion in paragraph 1, the proportionality relationship mentioned in paragraph 2 should be $T_{N2} \propto J_{c2} / J_{c1}$

Answers to Reviewer #3

We would like to thank the referee for its positive recommendation concerning the publication of the revised manuscript in Nature Communications.